# BRSET: A Brazilian Multilabel Ophthalmological Dataset of Retina Fundus Photos

**Luis Filipe Nakayama**[1,2]*, **David Restrepo**[2,3], **João Matos**[2,4], **Lucas Zago Ribeiro**[1], **Fernando Korn Malerbi**[1], **Leo Anthony Celi**[2,5,6], **Caio Saito Regatieri**[1]

**1** Department of Ophthalmology, São Paulo Federal University, São Paulo, São Paulo, Brazil, **2** Laboratory for Computational Physiology, Massachusetts Institute of Technology, Cambridge, Massachusetts, United States of America, **3** Telematics Department, University of Cauca, Popayán, Cauca, Colombia, **4** Faculty of Engineering of University of Porto, Porto, Portugal, **5** Division of Pulmonary, Critical Care and Sleep Medicine, Beth Israel Deaconess Medical Center, Boston, Massachusetts, United States of America, **6** Department of Biostatistics, Harvard T.H. Chan School of Public Health, Boston, Massachusetts, United States of America

* nakayama.luis@unifesp.br

**Data Availability Statement:** All the codes used in this paper for the dataset setup, data analysis, and

## Abstract

### Introduction

The Brazilian Multilabel Ophthalmological Dataset (BRSET) addresses the scarcity of publicly available ophthalmological datasets in Latin America. BRSET comprises 16,266 color fundus retinal photos from 8,524 Brazilian patients, aiming to enhance data representativeness, serving as a research and teaching tool. It contains sociodemographic information, enabling investigations into differential model performance across demographic groups.

### Methods

Data from three São Paulo outpatient centers yielded demographic and medical information from electronic records, including nationality, age, sex, clinical history, insulin use, and duration of diabetes diagnosis. A retinal specialist labeled images for anatomical features (optic disc, blood vessels, macula), quality control (focus, illumination, image field, artifacts), and pathologies (e.g., diabetic retinopathy). Diabetic retinopathy was graded using International Clinic Diabetic Retinopathy and Scottish Diabetic Retinopathy Grading. Validation used a ConvNext model trained during 50 epochs using a weighted cross entropy loss to avoid overfitting, with 70% training (20% validation), and 30% testing subsets. Performance metrics included area under the receiver operating curve (AUC) and Macro F1-score. Saliency maps were calculated for interpretability.

### Results

BRSET comprises 65.1% Canon CR2 and 34.9% Nikon NF5050 images. 61.8% of the patients are female, and the average age is 57.6 (± 18.26) years. Diabetic retinopathy affected 15.8% of patients, across a spectrum of disease severity. Anatomically, 20.2% showed abnormal optic discs, 4.9% abnormal blood vessels, and 28.8% abnormal macula.

experiments are found in a GitHub repository at https://github.com/luisnakayama/BRSET. The BRSET is available in https://physionet.org/content/brazilian-ophthalmological/1.0.0/.

**Funding:** The author(s) received no specific funding for this work.

**Competing interests:** The authors have declared that no competing interests exist.

A ConvNext V2 model was trained and evaluated BRSET in four prediction tasks: "binary diabetic retinopathy diagnosis (Normal vs Diabetic Retinopathy)" (AUC: 97, F1: 89); "3 class diabetic retinopathy diagnosis (Normal, Proliferative, Non-Proliferative)" (AUC: 97, F1: 82); "diabetes diagnosis" (AUC: 91, F1: 83); "sex classification" (AUC: 87, F1: 70).

## Discussion

BRSET is the first multilabel ophthalmological dataset in Brazil and Latin America. It provides an opportunity for investigating model biases by evaluating performance across demographic groups. The model performance of three prediction tasks demonstrates the value of the dataset for external validation and for teaching medical computer vision to learners in Latin America using locally relevant data sources.

### Author summary

In low-resource settings, access to open medical datasets is crucial for research. Regions such as Latin America often face underrepresentation, resulting in health biases and inequities. To face the scarcity of diverse ophthalmological datasets in these areas, especially in Brazil and Latin America, we introduce the Brazilian Multilabel Ophthalmological Dataset (BRSET) as a means to alleviate biases in medical AI research. Comprising 16,266 color fundus retinal photos from 8,524 Brazilian patients, BRSET integrates sociodemographic information, empowering researchers to investigate biases across demographic groups and diseases. BRSET was extracted from São Paulo outpatient centers, and includes demographics, clinical history, and retinal images labeled for anatomical features, quality control, and pathologies like diabetic retinopathy. Validation was performed in a set of selected prediction tasks, such as diabetes diagnosis, sex classification, and diabetic retinopathy diagnosis. BRSET's inclusion of sociodemographic data and experiment metrics underscores its potential efficacy across diverse classification objectives and patient groups, providing crucial insights for medical AI in underrepresented regions.

## Introduction

In ophthalmological practice, imaging assists in the diagnosis and follow-up of ocular conditions, including retinal fundus photos, ocular anterior segment photos, corneal topography, visual field tests, and optical coherence tomography [1,2]. Artificial intelligence (AI) algorithms can potentially improve medical care by facilitating access to screening, diagnosis, and monitoring in resource-limited settings and assist with the decision-making process [1–3]. In ophthalmology, AI holds promise for ocular diseases such as diabetic retinopathy, age-related macular degeneration, glaucoma, and retinopathy of prematurity [1,2,4–8]. While AI represents a breakthrough technology, unfair outcomes can arise from every step of the AI lifecycle, with models developed using non-representative and local data leading to unequal algorithms [9–12]. Among the sources of biases that can be induced in deep learning, those that stand out the most are those introduced by the data, the model, and the decision-making. A way to tackle biases introduced by data, is providing a representative open source dataset at both the sample size, as well as in knowledge through not only clinical variables but also demographic and sociocultural variables [13].

**Table 1. Comparative table with open-access ophthalmological datasets.**

| Dataset | View Position | Labels | Dataset Size | Image format | Annotation level | Retina specialist | Anatomical label | Diabetic retinopathy classification | Nationality |
|---|---|---|---|---|---|---|---|---|---|
| Eye Picture Archive Communication System [18] | Macular | Diabetic Retinopathy | 88702 images | JPG | Global | NA | No | ICDR | USA |
| BRSET | Macular | Multiple | 16266 images of 8524 patients | JPG | Global | Yes | Yes | ICDR and SDRG | Brazil |
| Jichi DR [19] | Macular | Diabetic Retinopathy | 9939 images | JPG | Global | NA | No | Davis | Japan |
| Asian Pacific Tele Ophthalmology society dataset [20] | Macular | Diabetic Retinopathy | 5593 images | PNG | Global | NA | No | ICDR | India |
| Ocular Disease Recognition (ODIR) [21] | Macular | Normal, diabetes, glaucoma, cataract, amd, hypertension, pathological myopia, other | 5000 images of 5000 patients | JPG | Global | NA | No | No | China |
| Retinal Fundus Multi-Disease Image Dataset (RFMiD): A Dataset for Multi-Disease Detection Research [22] | Macula and Optic disc | Multiple | 3200 images | PNG | Global | NA | No | No | India |
| Messidor 2 [23] | Macular | Diabetic Retinopathy | 1748 images | JPG, PNG | Global | No | No | No | France |
| DR1/DR2 [24] | Macular | Diabetic Retinopathy | 1597 images | TIFF | Segmentation | NA | No | No | Brazil |
| Rotterdam Ophthalmic data repository [25] | Macular and mosaic | Diabetic Retinopathy | 1120 images | PNG | Global | NA | No | No | Netherlands |
| Indian Diabetic Retinopathy Image Dataset [26] | Macular | Diabetic Retinopathy | 516 images | JPG | Global and Segmentation | Yes | No | ICDR | India |
| DIARETDB0 [27] | Macular | Diabetic Retinopathy | 130 images | PNG | Segmentation | NA | No | No | Finland |
| DIARETDB1 [28] | Macular | Diabetic Retinopathy | 89 images | PNG | Segmentation | Yes | No | No | Finland |
| E-ophtha [29] | Macular | Diabetic Retinopathy | 463 images | JPG | Segmentation | NA | No | No | France |
| Hamilton Eye Institute Macular Edema [30] | Macular | Diabetic Retinopathy | 169 images | JPG | Segmentation | Yes | No | No | USA |

The open science movement in healthcare has not gained traction in Latin America [14]. In ophthalmology, the majority of the available datasets come from high-income countries, as can be seen in Table 1. In addition, datasets lack demographic and crucial clinical information such as comorbidities [15]. In low and middle-income countries (LMIC), the number of ophthalmologists relative to the population is not adequate [16]. Autonomous systems may increase ophthalmological coverage and reduce preventable blindness; however, datasets that do not adequately represent those who are disproportionately impacted by the disease lead to biased and harmful algorithms. For instance, conditions like diabetic retinopathy and infectious retinopathies have a disproportionately higher prevalence in patients from LMICs, yet these individuals are often underrepresented in ophthalmological datasets [15,17].

The BRSET stands as a pioneering initiative within Latin America's ophthalmological landscape, offering a dataset that distinguishes itself by its multifaceted approach. While previous datasets have existed within the region, BRSET elevates the standard by encompassing a

diverse spectrum of diagnoses and sociodemographic categories. In essence, it marks a significant advancement as the inaugural repository to comprehensively integrate multiple diagnoses alongside sociodemographic parameters.

## Materials and methods

This study was approved by the Sao Paulo Federal University (UNIFESP) IRB (CAAE 33842220.7.0000.5505), and informed consent was waived due to efforts to ensure data anonymity, low risk of reidentification, and interruption of data sharing in any reidentification risk. Before accessing the dataset, the users must be credentialed, accept the Health Data use agreement, and complete training on data research. In this dataset, identifiable patient information was manually reviewed and removed from all images.

### Data sources

We included data from three outpatient Brazilian ophthalmological centers in São Paulo evaluated from 2010 to 2020 and from the Sao Paulo Federal University ophthalmology sector.

### Data collection

Images present in this dataset were collected using different retinal fundus cameras, including Nikon NF505 (Nikon, Tokyo, Japan), Canon CR-2 (Canon Inc, Melville, NY, USA) retinal camera, and Phelcom Eyer (Phelcom Technologies, MA, USA). Retinal photos were taken by previously trained non-medical professionals in pharmacological mydriasis.

### Dataset preparation

The file identification was removed from all fundus photos, as well as sensitive data (e.g., patient name, exam date). Every image was reviewed to ensure the absence of protected health information in images. The images were exported directly from retinal cameras in JPEG format, and no preprocessing techniques were performed. The image viewpoint can be macula-centered or optic disc-centered. The dataset does not include fluorescein angiogram photos, non-retinal images, or duplicated images.

### Metadata

Each retinal image is labeled with the retinal camera device, image center position, patient nationality, age in years, sex, comorbidities, insulin use, and duration of diabetes diagnosis. The demographics and medical features were collected from the electronic medical records.

### Labeling

A retinal specialist ophthalmologist labeled all the images based on criteria that were established by the research group [31]. The following characteristics were labeled:

- **Anatomic classification:** The retinal optic disc (vertical cup-disc ratio of $\geq 0.65$ [32]), retinal vessels (tortuosity and width), and macula (abnormal findings) were categorized as either normal or abnormal.

- **Quality control parameters:** Parameters including image focus, illumination, image field, and artifacts, were assessed and classified as satisfactory or unsatisfactory. The criteria are defined in Table 2.

**Table 2. Quality assessment parameters.**

| | |
|---|---|
| **Illumination** | This parameter is graded as adequate when both of the following requirements are met: 1) Absence of dark, bright, or washed-out areas that interfere with detailed grading; 2) In the case of peripheral shadows (e.g., due to pupillary constriction) the readable part should reach more than 80% of the whole image. |
| **Image Field** | This parameter is graded as adequate when all the following requirements are met: 1) The optic disc is at least 1 disc diameter (DD) from the nasal edge; 2) The macular center is at least 2 DD from the temporal edge; 3) The superior and inferior temporal arcades are visible in a length of at least 2 DD |
| **Artifacts** | The following artifacts are considered: haze, dust, and dirt. This parameter is graded as adequate when the image is sufficiently artifact-free to allow adequate grading. |
| **Focus** | This parameter is graded as adequate when the focus is sufficient to identify third-generation branches within one optic disc diameter around the macula. |

- **Pathological classifications:** The images were classified according to the pathological classification list: diabetic retinopathy, diabetic macular edema, scar (toxoplasmosis), nevus, age-related macular degeneration (AMD), vascular occlusion, hypertensive retinopathy, drusens, nondiabetic retinal hemorrhage, retinal detachment, myopic fundus, increased cup disc ratio, other.

- **Diabetic retinopathy classification:** Diabetic retinopathy was classified using the International Clinic Diabetic Retinopathy (ICDR) grading and Scottish Diabetic Retinopathy Grading (SDRG), as can be seen in Table 3.

## Data records

This dataset may be used to build computer vision models that predict demographic characteristics and multi-label disease classification. BRSET consists of 16,266 images from 8,524 Brazilian patients, and a metadata file. Columns are detailed in S1 File.

## Data storage

The dataset images and labels are stored on the PhysioNet repository entitled "A Brazilian Multilabel Ophthalmological Dataset (BRSET)". Link: https://physionet.org/content/brazilian-ophthalmological/1.0.0/.

**Table 3. Diabetic Retinopathy Classifications.**

| Classification | 0—Normal | 1—Mild non-proliferative DR | 2- Moderate non-proliferative DR | 3—Severe non-proliferative DR | 4—Proliferative DR | Macular edema |
|---|---|---|---|---|---|---|
| **International Classification of Diabetic Retinopathy** [33] | No abnormalities | Microaneurysms only | More than just microaneurysms but less than severe non-proliferative diabetic retinopathy | Any of the following: > 20 intra-retinal hemorrhages in each of 4 quadrants, definite venous beading in $\geq$2 quadrants, prominent intraretinal microvascular abnormalities in $\geq$1 quadrant, or no signs of proliferative retinopathy | One or more of the following: neovascularization and/or vitreous or preretinal hemorrhages and/or panfotocoagulation scars | Exudates or apparent thickening within one disc diameter from the fovea |
| **Scottish Diabetic Retinopathy Grading** [34] | No abnormalities | At least one microaneurysm, flame exudate, blot hemorrhage with or without hard exudate | More than 4 blot hemorrhages in one hemifield | More than 4 blot hemorrhages in both hemifields, IRMA, venous beading | Disc neovessels, retinal neovessels, vitreous hemorrhage, retinal detachment | Hard exudates within 1–2 DD of the fovea |

## Descriptive analytics and technical validation

**Descriptive analysis.** The BRSET database contains 10,592 (65.1%) images taken from a Canon CR2 retinal camera, and 5,674 (34.9%) are from the Nikon NF5050 retinal camera. The sex distributions in the dataset are 6,214 (38.2%) male patients, and 10,052 (61.8%) female patients. The average age is 57.6 years (standard deviation of 18.26 years).

The database includes 2,579 (15.8%) patients with a diagnosis of diabetes mellitus. Among those patients 1,922 (74.5%) do not have retinopathy, 107 (4.1%) have mild non-proliferative retinopathy, 181 (7%) have moderate non-proliferative retinopathy, 127 (4.9%) have severe non-proliferative retinopathy, and 242 (9.4%) have proliferative retinopathy. A sample of the images can be seen in Fig 1.

Among the anatomical criteria, 3,281 (20.2%) of the images have an abnormal optic disc, 807 (4.9%) have abnormal blood vessels, and 4,685 (28.8%) have an abnormal macula. The distribution of normal exams and pathological findings is described in Table 4.

## Data quality assessment

**Image quality.** In terms of quality criteria, 542 (3.3%) images were classified as having inadequate focus, 84 (0.5%) as having inadequate lighting, 1401 (8.6%) as having an inadequate field, and 57 (0.3%) as having artifacts.

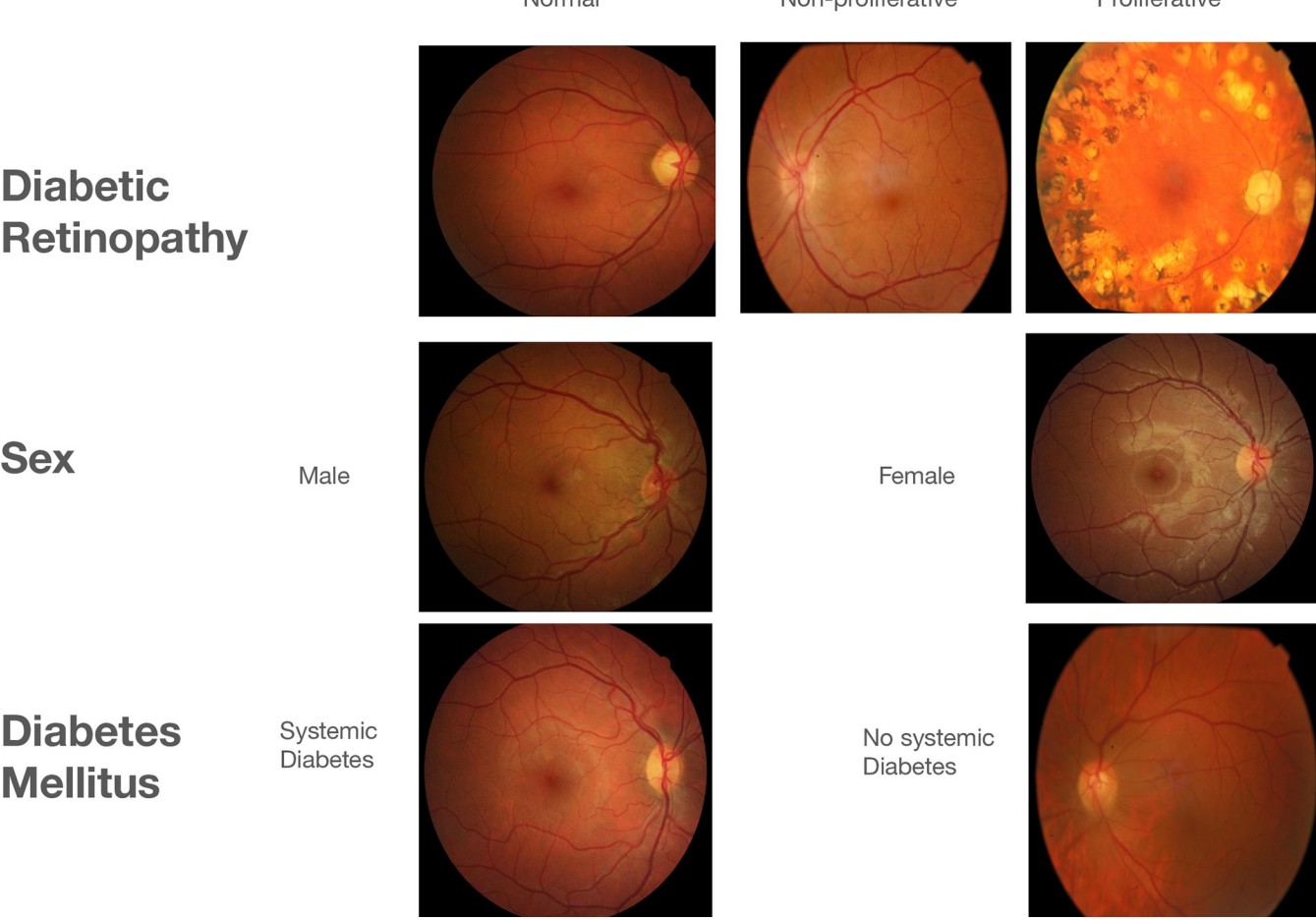

**Fig 1. Sample Retina Images with and without Diabetic Retinopathy (DR), male and female, and with systemic diabetes and without systemic diabetes, from the BRSET Dataset.**

**Table 4. Distribution of retina images according to diagnoses and anatomical criteria.**

| Labels | Images (n) | Percentage (%) |
|---|---|---|
| Normal | 8,460 | 52 |
| Diabetic retinopathy | 1,046 | 6.4 |
| Diabetic macular edema | 402 | 2.5 |
| Scar | 290 | 1.8 |
| Nevus | 134 | 0.8 |
| Age-related macular degeneration | 366 | 2.2 |
| Vascular occlusion | 103 | 0.6 |
| Hypertensive retinopathy | 283 | 1.7 |
| Drusens | 2,807 | 17.2 |
| Hemorrhage | 96 | 0.6 |
| Retinal detachment | 7 | 0.04 |
| Myopic fundus | 268 | 1.6 |
| Increased cup-disc ratio | 3,202 | 19.7 |
| Other | 758 | 4.7 |

**Quality metadata.** On the other hand, regarding the quality characteristics of the metadata, duplicate data and missing data were assessed. The dataset does not have duplicate data, and only 3 columns contain missing data: patient age column (33.47%), diabetes duration (88.26%), and insulin usage (89.46%). Notably, the latter two are exclusively recorded for diabetic patients.

**Experiments.** To explore the dataset, we provide use cases and benchmarks for future research. The codes required to run these experiments are available in a publicly available GitHub repository (see Code availability section).

In this section, we delineate the methodologies employed to validate the dataset proposed in our study. Our primary focus will be harnessing the application of ConvNext V2 as a deep learning method on the BRSET dataset for retina fundus photo analysis [35]. This investigation encompasses binary diabetic retinopathy classification, three-class diabetic retinopathy state classification (normal, non-proliferative, and proliferative retinopathy), diabetes mellitus prediction, and sex prediction.

As diabetic retinopathy is the leading cause of preventable blindness among adults and the most explored disease in publicly available datasets [15,17,36], we included diabetic retinopathy prediction and clinical diagnosis of diabetes mellitus through retina fundus photos.

To assess the ability to predict patient demographic characteristics through retinal fundus photos, extending beyond pathological findings, we included the prediction of patient sex, which is reported in previous studies [37,38].

**Model selection.** We opted for ConvNext V2, diverging from the conventional use of ResNet-50 used in prior research on the BRSET dataset [39–41]. The reason for using ConvNext V2 instead of ResNet-50, is that ConvNext V2 is an evolution of the ResNet-50 architecture, and introduces several enhancements over its predecessors, such as layer normalization, expanded kernel sizes, or regularization techniques [35,42]. The architecture is characterized by its depth-wise convolutional layers, which facilitate efficient feature extraction and scalability.

**Data preprocessing.** The BRSET dataset was stratified based on labels and divided into training (70%), validation (within training set: 20%), and testing sets (30%) using all the images available in the dataset. This stratification ensures a representative distribution of classes across each subset.

The images in the dataset present variations in the height, ranging from 874 to 2304 pixels; and the width, ranging from 951 to 2984 pixels. To ensure consistency with the model's input all the images were first resized to 256x256 from the original size, and randomly cropped to ensure a 224x224 shape during training. Also, all the images were normalized by dividing them into 225 to ensure all pixel values were between 0 and 1.

**Training strategy.** The model was trained using the full training set during 50 epochs incorporating an early stopping mechanism, keyed to F1 score improvements on the validation set, with a patience parameter set to 7 epochs. This implies that if no performance increase is observed over seven consecutive epochs, training ceases, reverting to the last optimal model state.

**Loss function and evaluation metrics.** All the models were trained for a classification task using a cross-entropy loss. Given the class imbalance inherent in the dataset, a weighted cross-entropy loss [43] function was employed using the formula in the Eq (1) [44], where c represents the number of classes, n represents the batch size, wc represents the weights of class c, and y and y hat represent the ground truth and predicted value respectively.

$$L(y, \hat{y}) = -\sum_{i=1}^{n} \sum_{c=1}^{c} w_c \cdot y_{i,c} \cdot log(\hat{y}_{i,c}) \tag{1}$$

Given that our predictive task is a classification task, the output of the model was defined using the number of classes as output neurons and using softmax activation function to return a set of probabilities for each class. The probability of each class was defined by the Eq (2) [45]; where z, is the logit corresponding to the class c of the i-th element in the batch. and $k$ corresponds to the number of possible classes.

$$\hat{y}_{i,c} = \frac{e^{z_{i,c}}}{\sum_{k=i}^{k} e^{z_{i,k}}} \tag{2}$$

Combining Eqs (1) and (2), we can see that the loss function of the models is given by Eq (3).

$$L(y, \hat{y}) = -\sum_{i=1}^{n} \sum_{c=1}^{c} w_c \cdot y_{i,c} \cdot log\left(\frac{e^{z_{i,c}}}{\sum_{k=i}^{k} e^{z_{i,k}}}\right) \tag{3}$$

The models were trained using class weights to tackle class imbalance during training. The weights were calculated using Eq (4), where w are the weights of class C, N is the total samples in the training dataset, K is the number of classes, and Nc is the samples of class in the training dataset.

$$w(C) = \frac{N}{K \cdot N_c} \tag{4}$$

Model performance was assessed using the Area Under the Receiver Operating Characteristic Curve (AUC-ROC), a performance metric used to evaluate the quality of predictions, extended to multi-class classification through macro-average.

The F1 score metric was also used to mitigate bias due to class imbalance. The F1 score can be seen in Eq (7) and is a harmonic mean of precision (5) and recall (6) [46]. For the multiclass problems, the macro average was applied to the F1 score to avoid biased results.

$$Precision = \frac{TP}{TP + FP} \tag{5}$$

$$Recall = \frac{TP}{TP + FN} \tag{6}$$

$$F1 = 2 \cdot \frac{Precision \cdot Recall}{Precision + Recall} \tag{7}$$

**Optimization and interpretability.** Adam optimizer [47] was utilized with a learning rate of 1e-5 for diabetes prediction, binary diabetic retinopathy, and 3-class diabetic retinopathy; and 5e-6 was used for sex prediction, over a maximum of 50 epochs or until early stopping was triggered. To enhance model clinical interpretability, we generated saliency maps for select images across classes [48]. Saliency maps highlight regions within an image most influential to the model's decision, computed by taking the gradient of the output with respect to the input image. Saliency maps are calculated using Eq 8 [48], where S(x) denotes the saliency map, Y is the model output, and X is the input image. This provides insights into model focus areas and potential biases.

$$S(x) = \frac{\partial y}{\partial x} \tag{8}$$

## Results

The efficacy of ConvNext V2 was benchmarked against state-of-the-art (SOTA) models documented in previous studies utilizing the same dataset. This comparison extends to saliency maps, offering a visual and quantitative analysis of model attention in diagnosing diabetic retinopathy and determining sex, underscoring the model's practical applicability in real-world settings. (Table 5)

The task of binary diabetic retinopathy classification achieved an AUC of 0.97 and a Macro F1-score of 0.89 with ConvNext V2, matching the AUC but significantly surpassing the F1-score of ResNet 50. This improvement in the F1-score, a measure sensitive to class imbalance, underscores the effectiveness of ConvNext V2 in balancing sensitivity and specificity, a crucial aspect in clinical diagnostics.

The three-class diabetic retinopathy diagnosis further emphasizes the model's robustness, with ConvNext V2 achieving an AUC of 0.97 and a Macro F1-score of 0.82, establishing comparable results with the state-of-the-art (SOTA) results for the BRSET dataset.

The classification of diabetes mellitus, a more challenging task due to the subtler manifestation of the disease in retina images, saw a commendable performance with an AUC of 0.87 and a Macro F1-score of 0.70. This underscores the ConvNext V2's capability to extract nuanced features indicative of early or less pronounced disease states, which are critical for early intervention and management.

**Table 5. Performance metrics of the BRSET for the four selected classification tasks.** The tasks are binary diabetic retinopathy classification, three-class diabetic retinopathy state classification (normal, non-proliferative, and proliferative), diabetes mellitus prediction, and sex prediction.

| Task | Model | AUC | Macro F1-score |
|---|---|---|---|
| Binary diabetic retinopathy diagnosis (Normal vs any Diabetic Retinopathy) | ConvNext V2 Large (ours) | 0.97 | 0.89 |
| | ResNet 50 [41] | 0.97 | 0.82 |
| 3-class diabetic retinopathy diagnosis (Normal vs Non-Proliferative vs Proliferative) | ConvNext V2 Large (ours) | 0.97 | 0.82 |
| | ResNet-200D [39] | 0.95 | 0.83 |
| Sex classification | ConvNext V2 Large (ours) | 0.91 | 0.83 |
| | ResNet-200D [39] | 0.80 | NA |
| Diabetes Mellitus Classification | ConvNext V2 Large (ours) | 0.87 | 0.70 |

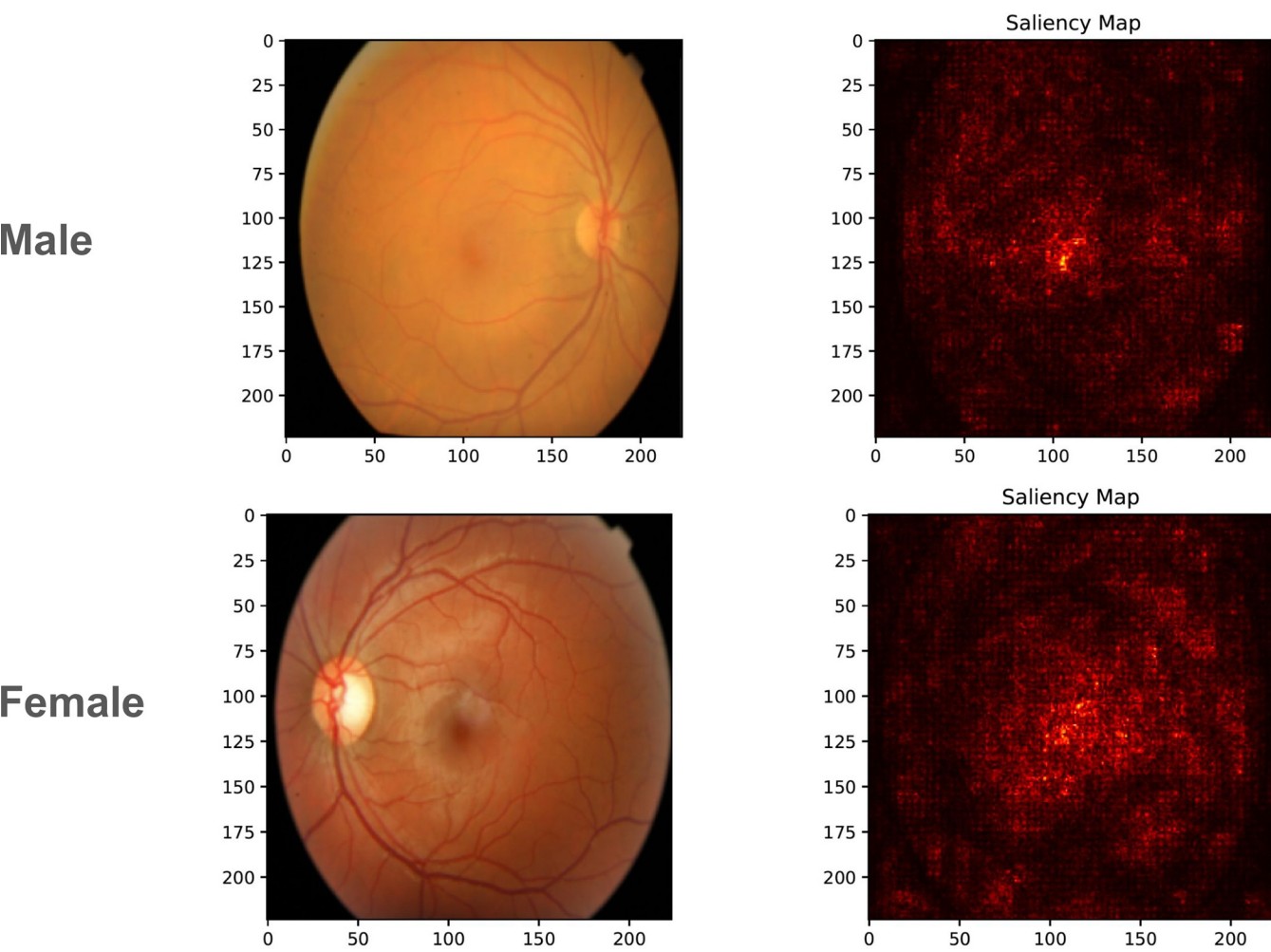

**Fig 2. Saliency maps comparing the model's focus areas.** For sex prediction, the model assesses more global retinal features.

In the sex classification task, the model achieved an AUC of 0.91 and a Macro F1-score of 0.83, demonstrating the model's ability to discern patterns beyond pathological features, potentially leveraging anatomical and physiological differences captured in retina images.

Saliency maps have emerged as a pivotal tool in interpreting deep learning models, particularly in medical imaging, by highlighting regions within an image that significantly influence the model's decision. The maps in Fig 2 and Fig 3 reveal that, for tasks such as sex prediction, ConvNext V2 tends to focus on more global features of the retina, whereas, for diabetic retinopathy classification, the model's attention is drawn to more localized features indicative of the disease, such as microaneurysms, hemorrhages, and retinal exudates.

## Discussion

The Brazilian Multilabel Ophthalmological Dataset (BRSET) represents a significant stride towards mitigating biases in healthcare research. The inclusion of a comprehensive dataset encompassing of Brazilian patients, coupled with extensive metadata, provides a robust foundation for enhancing data representativeness in medical AI applications. This dataset not only facilitates the exploration of models by evaluating performance across diverse demographic

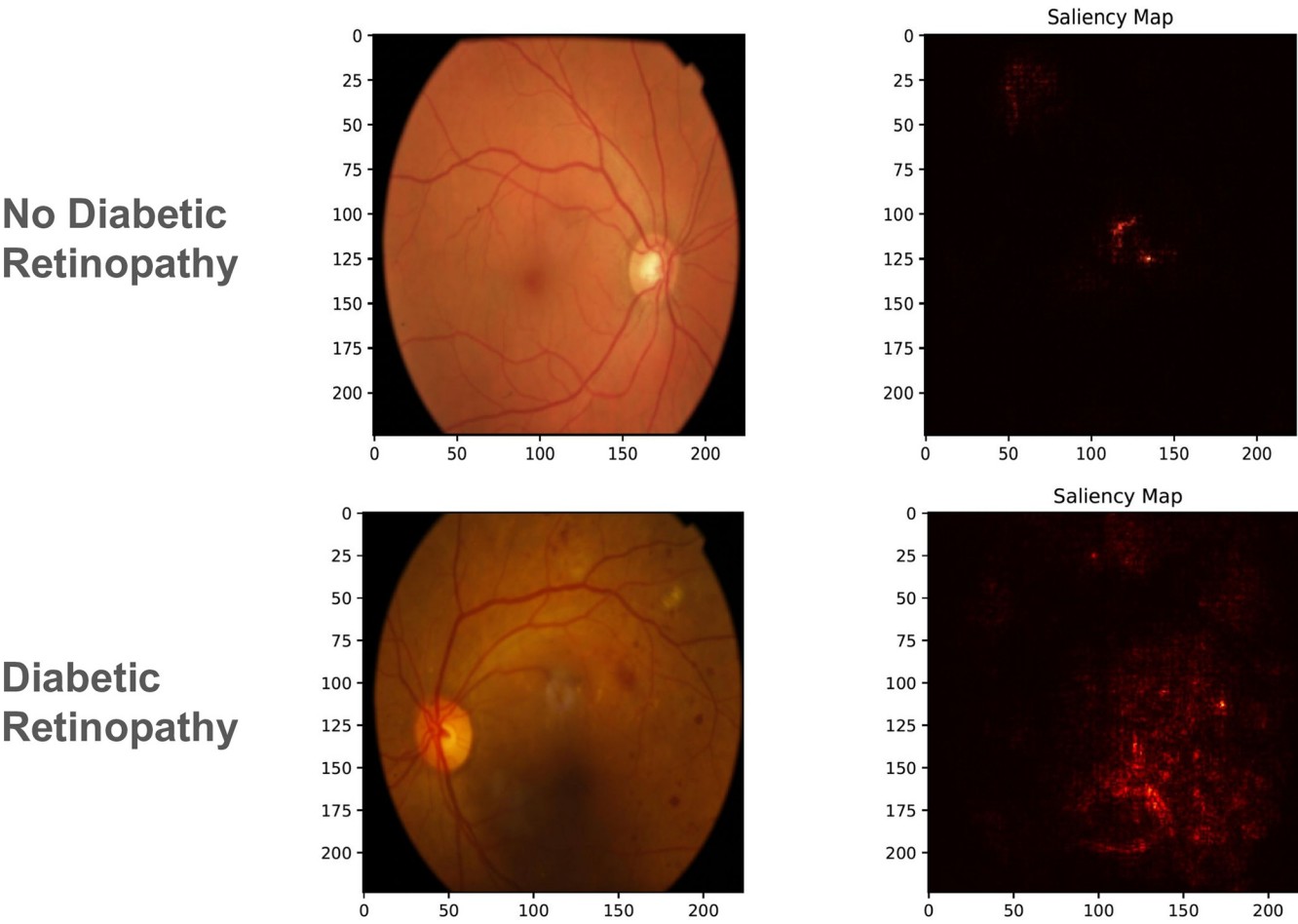

**Fig 3. Saliency maps comparing the model's focus areas.** For diabetic retinopathy, localized regions indicative of pathology are highlighted.

groups but also serves as a valuable resource for teaching medical computer vision in regions where relevant local data is scarce.

The utility of the BRSET extends beyond mere data representation; it serves as a critical tool in addressing biases prevalent in healthcare AI, particularly in underrepresented populations. The integration of sociodemographic information enables a nuanced investigation into differential model performance, offering insights into potential biases and disparities. This aspect is crucial for Latin America and similar regions, where healthcare disparities are pronounced, and data representation in medical research is limited [14].

The employment of advanced deep learning techniques, such as ConvNext V2, further underscores the dataset's utility [35] and has yielded notable advancements in the field of retina fundus photo analysis. The results showcase the superiority of the architecture over the previously established benchmarks, primarily those set by ResNet 50 in the classification tasks of binary diabetic retinopathy and three-class diabetic retinopathy states [39,41].

Notably, the ConvNext V2 model's ability to discern subtle patterns in retinal images, as evidenced by its commendable performance in diabetes mellitus classification and sex prediction, highlights the dataset's versatility, with results comparable to previous studies [37,38].

Interpretability methods, such as saliency maps, play a pivotal role in demystifying the decision-making process of AI models for clinical practitioners. In our study, saliency maps

revealed the model's focus areas, distinguishing between localized pathological features for diabetic retinopathy and more global retinal features for sex prediction, corroborating the human clinical classification. This level of interpretability is vital for clinical acceptance, allowing healthcare professionals to understand and trust AI-assisted diagnostics.

## Usage notes

BRSET is available on PhysioNet as a database that requires user credentialing prior to access. Users must be registered on PhysioNet, have proper human subject research training, and sign a data use agreement that forbids re-identification of patients and sharing it with those who are not credentialed.

## Limitations

Our dataset only includes one nationality and represents general ophthalmological outpatients. As a result, the disease distribution is imbalanced, with a high percentage of normal and mild cases (Table 4). Those limitations may restrict the generalizability of findings to broader Latin American populations. This limitation underscores the necessity for future research to expand the dataset, incorporating data from various Latin American countries to enhance its representativeness and utility.

A single expert evaluation risks introducing bias and undermining the credibility of labels. To enhance the robustness of the BRSET, our future steps target employing two ophthalmologists for labeling, supplemented by adjudication from either a retina specialist or a glaucoma specialist.

This project sought to address the issue of underrepresentation in ophthalmological data from Brazil and Latin America. However, biases in AI are complex and multifaceted. While improving data representativeness is a crucial step, it is only one aspect among many that needs consideration.

Our technical validation results are biased towards the distribution of age, sex, race, among many others in our dataset. Future work should analyze and address different types of social biases included in our dataset and during model development as well as the use of models, methods such as policy loss function, and model deployment. We highly encourage the community to expand our analysis over our dataset and models to detect and reduce other sources of biases.

## Strengths and conclusions

The BRSET is the first multilabel ophthalmological dataset from Brazil and Latin America. BRSET aims to improve data representativeness and create the framework for ophthalmological dataset development. To the best of our knowledge, BRSET is the only publicly available retinal image dataset that also contains sociodemographic data such as sex and age in Latin America, which allows us to investigate algorithmic bias across different demographic groups.

The superior performance of ConvNext V2, demonstrated through enhanced AUC and Macro F1-scores across multiple classification tasks on the BRSET dataset, underscores its suitability for clinical applications. The model not only sets new benchmarks but also provides a deeper understanding of disease markers through interpretability tools like saliency maps, bridging the gap between AI models and clinical applicability. These insights can guide clinicians in decision-making and foster trust in AI-assisted diagnostics, marking a significant leap forward in the integration of deep learning technologies in healthcare.

### Release note

We plan to include self-declared race and demographic data and increase the ophthalmological exam modalities for future releases.

## Supporting information

**S1 File. Data Dictionary–description of BRSET columns.**
(DOCX)

## Author Contributions

**Conceptualization:** Luis Filipe Nakayama, Lucas Zago Ribeiro, Fernando Korn Malerbi, Caio Saito Regatieri.

**Data curation:** Luis Filipe Nakayama, David Restrepo, João Matos, Lucas Zago Ribeiro, Fernando Korn Malerbi.

**Formal analysis:** Luis Filipe Nakayama, David Restrepo.

**Investigation:** Luis Filipe Nakayama, Lucas Zago Ribeiro.

**Methodology:** Luis Filipe Nakayama, David Restrepo.

**Project administration:** Luis Filipe Nakayama.

**Supervision:** Leo Anthony Celi, Caio Saito Regatieri.

**Validation:** João Matos, Fernando Korn Malerbi, Leo Anthony Celi.

**Visualization:** Luis Filipe Nakayama, David Restrepo, João Matos.

**Writing – original draft:** Luis Filipe Nakayama, David Restrepo, João Matos, Lucas Zago Ribeiro, Fernando Korn Malerbi, Leo Anthony Celi, Caio Saito Regatieri.

**Writing – review & editing:** Luis Filipe Nakayama, David Restrepo, João Matos, Lucas Zago Ribeiro, Fernando Korn Malerbi, Leo Anthony Celi, Caio Saito Regatieri.

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
