## [Decision Letter · Decision Letter 0]

14 Feb 2024

PDIG-D-24-00024

BRSET: A Brazilian Multilabel Ophthalmological Dataset of Retina Fundus Photos

PLOS Digital Health

Dear Dr. Nakayama,

Thank you for submitting your manuscript to PLOS Digital Health. After careful consideration, we feel that it has merit but does not fully meet PLOS Digital Health's publication criteria as it currently stands. Therefore, we invite you to submit a revised version of the manuscript that addresses the points raised during the review process.

Please submit your revised manuscript within 60 days Apr 14 2024 11:59PM. If you will need more time than this to complete your revisions, please reply to this message or contact the journal office at digitalhealth@plos.org. Please include the following items when submitting your revised manuscript:

We look forward to receiving your revised manuscript.

Kind regards,

Miguel Ángel Armengol de la Hoz, Ph.D.

Section Editor

PLOS Digital Health

Journal Requirements:

1. Please provide separate figure files in .tif or .eps format only and remove any figures embedded in your manuscript file. Please also ensure all files are under our size limit of 10MB.

Additional Editor Comments (if provided):

Reviewers' comments:

Reviewer's Responses to Questions

**Comments to the Author**

1. Does this manuscript meet PLOS Digital Health’s publication criteria? Is the manuscript technically sound, and do the data support the conclusions? The manuscript must describe methodologically and ethically rigorous research with conclusions that are appropriately drawn based on the data presented.

Reviewer #1: No

Reviewer #2: Partly

Reviewer #3: Yes

2. Has the statistical analysis been performed appropriately and rigorously?

Reviewer #1: No

Reviewer #2: No

Reviewer #3: N/A

3. Have the authors made all data underlying the findings in their manuscript fully available (please refer to the Data Availability Statement at the start of the manuscript PDF file)?

Reviewer #1: Yes

Reviewer #2: Yes

Reviewer #3: Yes

4. Is the manuscript presented in an intelligible fashion and written in standard English?

Reviewer #1: Yes

Reviewer #2: Yes

Reviewer #3: Yes

5. Review Comments to the Author

Reviewer #1: Nakayama et. al. published BRSET, an multilevel ophthalmological dataset containing data from the unrepresented Brazil and Latin America population. They performed an exploratory data analysis on BRSET and propose the use of machine learning algorithms for disease and sex classification. 

The data curation and availability of the BRSET dataset is of high importance. However the authors failed to demonstrate the novelty in their approach or draw conclusions from the results driven. The data analysis presented is superficial. The results presented by ML algorithms are average and almost random - F1 and AUC close to 0.6 is not much better than by chance.

The method descriptions need to be substantially developed. The choice of the feature extraction approach using Dino V2 needs to be addressed. The authors did not explain how they went from an input matrix image to a feature vector with size 768 using Dino V2. 

In fact, what are the initial image sizes? Are all images the same size? Was any pre-processing employed? Data description and preprocessing needs to be addressed.

The authors mention that they are the only ophthalmological dataset from Latin America and Brazil, however they present in Table 1 a brazilian dataset named DR1/DR2. The authors should say how BRSET is different from this data. There is no reference for DR1/DR2 dataset, or any other dataset in this table.

The authors commented on the quality of the images, where around 12.2% presented some quality problem. However, they do not inform what was done with these images. Were they used when training the ML algorithms? How were the quality issues addressed?

Authors used SVC and LR to classify diseases and patient sex. However, there are several deep learning approaches that looks more suitable and showed better performance such as the ones presented in references [1, 2]. Why did the authors decided to employ these techniques that underperformed when compared with the literature? 

Readability suggestions:

- Table 4 does not seem necessary and the available features can be easily described in the text or in a supplement material.

- Simple average and std dev for age feature can summarize Fig 1.

- Fig 3 can be described directly in the text. Authors should explain how they handle missing data.

- There are no comments or description for Figures and Tables. If a figure or table is inserted in the paper, it should be presented and commented on and conclusions drawn, when applicable.

- A figure describing the proposed framework would help understanding the paper’s goal. 

- “Dino V2 Base [18]” should be reference [20]

- “as can be seen in Table 3 [18,19].” Reference should be updated accordingly.

Reviewer #2: I have two major concerns:

1) Ethical: The authors should not waive themselves for not having the consent from the subjects from whom the retina picture were used. The fact that the pictures were anonymised is not suffisent. The waiver should be given by an Independant Review Commity.

2) Adjudication of the diagnostic and other data: it is not sufficient that a single expert review all the pictures. It should be at least 2 independant experts with a mechanism for resolving disagrements. This would add significantly on the credibility of the data.

Reviewer #3: In this work, the authors present the BRSET database, which includes information from retinal photographs of 8,524 Brazilian patients, including sociodemographic and clinical information labeled by a retina specialist. This data is available on PhysioNet (although not referenced as required by the site). The authors detail the dataset and the different labeling stages comprehensively. Additionally, the authors conduct classification experiments using classic machine learning techniques such as SVM and LR to diagnose diabetes, classify by sex, or diagnose diabetic retinopathy. The document is clearly written, but I have some observations that, in my opinion, need to be addressed before publication.

Major:

- I understand that the experiments have been used as a basis; however, it is not clear why these tasks were chosen. I suggest incorporating a proper justification based on the literature of why these tasks are relevant for this database.

- Considering the amount of data I deem sufficient, the results do not seem to be promising. I would ask the authors to provide their insight regarding the results compared with the state of the art. Are these metrics common in other types of datasets? Is there information that this experiment is providing that contributes to the differentiation between classes of the datasets?

Both comments could be included in a discussion paragraph.

Minor:

- Add the standard deviation of age in the abstract.

- Include information about the results of the machine learning techniques in the abstract.

- More examples could be added in Figure 2 (or additional figures) that contribute to the differentiation between different classes for various tasks. These visual differences would better contribute to the discussion not only of the article but also of future research.

6. PLOS authors have the option to publish the peer review history of their article (what does this mean?). If published, this will include your full peer review and any attached files.

**Do you want your identity to be public for this peer review?** For information about this choice, including consent withdrawal, please see our Privacy Policy.

Reviewer #1: No

Reviewer #2: Yes: Dr Vincent Grek

Reviewer #3: No

---

## [Decision Letter · Decision Letter 1]

26 Mar 2024

PDIG-D-24-00024R1

BRSET: A Brazilian Multilabel Ophthalmological Dataset of Retina Fundus Photos

PLOS Digital Health

Dear Dr. Nakayama,

Thank you for submitting your manuscript to PLOS Digital Health. After careful consideration, we feel that it has merit but does not fully meet PLOS Digital Health's publication criteria as it currently stands. Therefore, we invite you to submit a revised version of the manuscript that addresses the points raised during the review process.

Please submit your revised manuscript within 30 days Apr 25 2024 11:59PM. If you will need more time than this to complete your revisions, please reply to this message or contact the journal office at digitalhealth@plos.org. Please include the following items when submitting your revised manuscript:

We look forward to receiving your revised manuscript.

Kind regards,

Miguel Ángel Armengol de la Hoz, Ph.D.

Section Editor

PLOS Digital Health

Journal Requirements:

Additional Editor Comments (if provided):

Reviewers' comments:

Reviewer's Responses to Questions

**Comments to the Author**

1. If the authors have adequately addressed your comments raised in a previous round of review and you feel that this manuscript is now acceptable for publication, you may indicate that here to bypass the “Comments to the Author” section, enter your conflict of interest statement in the “Confidential to Editor” section, and submit your "Accept" recommendation.

Reviewer #4: All comments have been addressed

Reviewer #5: All comments have been addressed

Reviewer #6: All comments have been addressed

Reviewer #7: All comments have been addressed

2. Does this manuscript meet PLOS Digital Health’s publication criteria? Is the manuscript technically sound, and do the data support the conclusions? The manuscript must describe methodologically and ethically rigorous research with conclusions that are appropriately drawn based on the data presented.

Reviewer #4: Yes

Reviewer #5: Yes

Reviewer #6: Yes

Reviewer #7: Partly

3. Has the statistical analysis been performed appropriately and rigorously?

Reviewer #4: Yes

Reviewer #5: Yes

Reviewer #6: Yes

Reviewer #7: Yes

4. Have the authors made all data underlying the findings in their manuscript fully available (please refer to the Data Availability Statement at the start of the manuscript PDF file)?

Reviewer #4: Yes

Reviewer #5: Yes

Reviewer #6: Yes

Reviewer #7: Yes

5. Is the manuscript presented in an intelligible fashion and written in standard English?

Reviewer #4: Yes

Reviewer #5: Yes

Reviewer #6: Yes

Reviewer #7: No

6. Review Comments to the Author

Reviewer #4: Based on the reviews provided to your paper, there are some points that can be amended or clarified:

1. Additional examples can be added in Figure 2 or a new format can be added to facilitate differentiation between different categories in multiple tasks.

2. More details about the selection of proposed tasks and experiments can be clarified and the feasibility of these choices can be emphasized based on scientific literature.

3. Results from machine learning techniques can be improved and explained and compared to normal performance in other types of data sets.

These points can be addressed to improve the quality of the paper and better clarify the content.

Reviewer #5: Many thanks for the opportunity to review this important manuscript. Addressing bias in the development and use of artificial intelligence is of vital importance.

I acknowledge that the previous review’s suggestions and feedback have been addressed within this revised manuscript. 

I do however have some suggestions for further improvement.

1. “concerns for unfair algorithms resulting from non-representative data and biased models cannot be ignored”

The representativeness of data set is one of a number of biases that can be introduced during the development and use of AI - it would strengthen this paper if the authors acknowledged this and noting whether the authors sought to consider and address other sources of bias. What residual biases might remain in the model?

2. “datasets that do not adequately represent those who are disproportionately impacted by the disease lead to biased and harmful algorithms” 

 I suggest the authors provide examples to demonstrate this point.

3."The Brazilian Multilabel Ophthalmological Dataset (BRSET) represents a significant stride towards mitigating biases in healthcare research"

The authors do not mention whether the model was evaluated post training to provide assurance that it was performing as intended and adequately addressing bias? If not, this would be a limitation and the authors might want to suggest that further work on this might be required as well as addressing other sources of bias in AI development.

Reviewer #6: This is a very valuable contribution to the literature. The authors have adequately addressed all of the reviewer comments and have made additional changes in accordance with the reviewer comments. The manuscript is quite improved as a result.

An aside, the use and discussion of saliency maps in this context is particularly valuable to the larger discussion on interpretability and explainability and to the larger discussion on equity in the integration of AI in healthcare.

Reviewer #7: This this work, the authors present the BRSET database made publicly available on PhysioNet. This include the information of 8 524 Brazilian patients, including sociodemographic and clinical information labeled by just one retina specialist. The authors addressed most of the comments and suggestions given in the past and the improvements are noticeable, mainly in the performance of the presented models. Also added most of the limitations of this work in the conclusions. The document is clearly written, but there are some sections that do not have the appropriate tone to sound polished and professional. In my opinion, this work needs to address the following points.

Major:

- As mentioned by Reviewer # 2, there is big ethical concern regarding the data. They provided a comprehensible response to this reviewer, but this is not commented in the main text. Do the authors had the permissions to bring out of Brazil to be published in a server that may be located in the US? What does the Brazilian data laws (if exist) recommend?

- I agree with reviewer #2 and being that you may not be able to re-run the labelling with another retina expert, you gave that response that is understandable. However, what about when these 2 clinicians provide different diagnosis/label? How do you decide which is the right label? Maybe this should be considered for future iterations.

- References. Even though this is not a PhD thesis, there are some parameters that should be complied. There are a lot of methods and equations mentioned in the text that do not have the appropriate reference. For instance, cross-entropy loss (line 255, pp 14) where the authors took this equation or method? Where does another student will find this information if they want to apply the same method? This comments applies to all equations (1 - 6). Adam optimizer (pp 15). Softmax (pp 14). Salience maps (pp 17), who said that?

- Interpretability. For whom is interpretable your findings. This is not clear in the text. As a Data Scientist, it is not completely clear.

Minor:

- Why do you add equtiions that are not referenced in the main text? Precision and recall equations, above equation 5.

- I am afraid that the font style is not appropriate to reference equation variables or other mathematical variables in the main text. Normal font instead of italic?

- Limitations (pp 19). Why don’t you mention (Again) the normal and mild cases? By this time the reader already forgot these values. At least, refer to the table that contains that information.

- There are some contractions that I do not find appropriate in an scientific work in a high profile journal: We’ll …. There’s. Correct them please.

7. PLOS authors have the option to publish the peer review history of their article (what does this mean?). If published, this will include your full peer review and any attached files.

**Do you want your identity to be public for this peer review?** For information about this choice, including consent withdrawal, please see our Privacy Policy. 

Reviewer #4: Yes: Noor aldeen mohammad yousefshehab

Reviewer #5: Yes: Shoshana Bloom

Reviewer #6: No

Reviewer #7: No

---

## [Decision Letter · Decision Letter 2]

3 Jun 2024

BRSET: A Brazilian Multilabel Ophthalmological Dataset of Retina Fundus Photos

PDIG-D-24-00024R2

Dear Dr. Nakayama,

We are pleased to inform you that your manuscript 'BRSET: A Brazilian Multilabel Ophthalmological Dataset of Retina Fundus Photos' has been provisionally accepted for publication in PLOS Digital Health.

Best regards,

Miguel Ángel Armengol de la Hoz, Ph.D.

Section Editor

PLOS Digital Health

Reviewer Comments (if any, and for reference):

Reviewer's Responses to Questions

**Comments to the Author**

1. If the authors have adequately addressed your comments raised in a previous round of review and you feel that this manuscript is now acceptable for publication, you may indicate that here to bypass the “Comments to the Author” section, enter your conflict of interest statement in the “Confidential to Editor” section, and submit your "Accept" recommendation.

Reviewer #8: All comments have been addressed

Reviewer #9: All comments have been addressed

2. Does this manuscript meet PLOS Digital Health’s publication criteria? Is the manuscript technically sound, and do the data support the conclusions? The manuscript must describe methodologically and ethically rigorous research with conclusions that are appropriately drawn based on the data presented.

Reviewer #8: Yes

Reviewer #9: Yes

3. Has the statistical analysis been performed appropriately and rigorously?

Reviewer #8: N/A

Reviewer #9: Yes

4. Have the authors made all data underlying the findings in their manuscript fully available (please refer to the Data Availability Statement at the start of the manuscript PDF file)?

Reviewer #8: No

Reviewer #9: Yes

5. Is the manuscript presented in an intelligible fashion and written in standard English?

Reviewer #8: Yes

Reviewer #9: Yes

6. Review Comments to the Author

Reviewer #8: The manuscript on the BRSET dataset and ConvNext V2 model is comprehensive and well-structured, making a valuable contribution to medical AI research. However, to meet journal publication standards, core improvements are needed, particularly in expanding the discussion on data representativeness and addressing potential limitations.

Here are some suggestions for core improvements to enhance the manuscript's suitability for publication in a journal:

1. Dataset Description: Expand the section describing the BRSET dataset. Provide more details on the dataset's creation, including the selection criteria for images, the data collection process, and any quality control measures implemented. This will help readers better understand the dataset's characteristics and potential applications.

2. Data Quality Assessment: Provide a more comprehensive analysis of data quality assessment. Discuss the implications of missing data in the patient age, diabetes duration, and insulin usage columns on the dataset's usability and potential biases. Consider proposing strategies to mitigate these issues or acknowledging their impact on the study's findings.

3. Model Selection and Justification: Further justify the selection of ConvNext V2 over other deep learning architectures, such as ResNet-50. Explain how ConvNext V2's specific features and enhancements make it well-suited for analyzing retina fundus photos in the context of the BRSET dataset. This will strengthen the rationale behind the model choice and enhance the manuscript's scientific rigor.

4. Technical Validation: Provide additional details on the technical validation methodologies employed in the study. Discuss any validation techniques used to assess the robustness and generalizability of the ConvNext V2 model across different subsets of the BRSET dataset. This will increase transparency and help validate the model's performance in real-world scenarios.

5. Discussion of Sociodemographic Factors: Incorporate a discussion on how sociodemographic factors, such as age, sex, and ethnicity, may influence the model's performance and the dataset's representativeness. Consider analyzing whether there are any disparities in model performance across different demographic groups and discuss potential implications for healthcare equity and access.

6. Ethical Considerations: Address ethical considerations related to the use of patient data in the BRSET dataset. Discuss how patient privacy and confidentiality were maintained throughout the data collection and analysis process. Provide information on any institutional review board approvals or ethical clearances obtained for the study.

By addressing these core areas for improvement, the manuscript will be better positioned to provide valuable insights into the development and application of the BRSET dataset in the field of ophthalmology and medical AI.

Reviewer #9: Dear Authors

Thank you for taking the time to adequately respond to the comments raised from the previous reviews. Thank you for also taking on this initiative of creating a “novel” A Brazilian Multilabel Ophthalmological Dataset of Retina Fundus Photos. This will promote data science research and innovations. I hope you can share your datasets through several platforms like Kaggle, and PhysioNet which you have already done, and then conduct several challenges to promote the utilisation of this dataset.

7. PLOS authors have the option to publish the peer review history of their article (what does this mean?). If published, this will include your full peer review and any attached files.

**Do you want your identity to be public for this peer review?** For information about this choice, including consent withdrawal, please see our Privacy Policy.

Reviewer #8: No

Reviewer #9: **Yes: **Wasswa William
